# Synthesis of Spherical Nanoparticle Hybrids via Aerosol Thiol-Ene Photopolymerization and Their Bioconjugation

**DOI:** 10.3390/nano12030577

**Published:** 2022-02-08

**Authors:** Narmin Suvarli, Max Frentzel, Jürgen Hubbuch, Iris Perner-Nochta, Michael Wörner

**Affiliations:** Department of Bio- and Chemical Engineering, Institute of Process Engineering in Life Sciences, Section IV: Biomolecular Separation Engineering, Karlsruhe Institute of Technology, 76131 Karlsruhe, Germany; narmin.suvarli@kit.edu (N.S.); max.frentzel@gmx.net (M.F.); juergen.hubbuch@kit.edu (J.H.); iris.perner-nochta@kit.edu (I.P.-N.)

**Keywords:** nanoparticle hybrids, silver stabilization, thiol-ene polymerization, aerosol photopolymerization, thiol-maleimide bioconjugation, streptavidin-biotin-binding

## Abstract

Hybrid nanomaterials possess the properties of both organic and inorganic components and find applications in various fields of research and technology. In this study, aerosol photopolymerization is used in combination with thiol-ene chemistry to produce silver poly(thio-ether) hybrid nanospheres. In aerosol photopolymerization, a spray solution of monomers is atomized, forming a droplet aerosol, which then polymerizes, producing spherical polymer nanoparticles. To produce silver poly(thio-ether) hybrids, silver nanoparticles were introduced to the spray solution. Diverse methods of stabilization were used to produce stable dispersions of silver nanoparticles to prevent their agglomeration before the photopolymerization process. Successfully stabilized silver nanoparticle dispersion in the spray solution subsequently formed nanocomposites with non-agglomerated silver nanoparticles inside the polymer matrix. Nanocomposite particles were analyzed via scanning and transmission electron microscopy to study the degree of agglomeration of silver nanoparticles and their location inside the polymer spheres. The nanoparticle hybrids were then introduced onto various biofunctionalization reactions. A two-step bioconjugation process was developed involving the hybrid nanoparticles: (1) conjugation of (biotin)-maleimide to thiol-groups on the polymer network of the hybrids, and (2) biotin-streptavidin binding. The biofunctionalization with gold-nanoparticle-conjugates was carried out to confirm the reactivity of -SH groups on each conjugation step. Fluorescence-labeled biomolecules were conjugated to the spherical nanoparticle hybrids (applying the two-step bioconjugation process) verified by Fluorescence Spectroscopy and Fluorescence Microscopy. The presented research offers an effective method of synthesis of smart systems that can further be used in biosensors and various other biomedical applications.

## 1. Introduction

Development of new sustainable eco-efficient [1,2] methods for synthesis of nanomaterials has become a focus of many researchers in the last decades [3]. Synthesis of organic-inorganic hybrid nanoparticles can be carried out using various techniques [4,5]. In situ free radical chain polymerization in bulk is one of the widely used industrial methods to produce nanocomposites [6,7,8]. Harmful matrices, capping agents, and toxic solvents in the synthesis of organic-inorganic nanocomposites are used very often [9]. Aerosol photopolymerization offers an environment-friendly, cost-effective, continuous flow-through method for the synthesis of nanoparticles [10,11,12]. In contrast to other polymerization methods, aerosol photopolymerization does not require heating, high-energy homogenization, surfactants, stabilizers, and co-stabilizers. Aerosol photopolymerization has already been used to synthesize zinc oxide-polymer nanoparticle hybrids [11], and was chosen in this study because the photopolymerization reaction is supported by the instantaneous formation of radicals and presents a more effective and simple method over thermally initiated polymerization, thus, the synthesis can be carried out at room temperature in an integrated continuous process.

Promising studies have demonstrated extraordinary benefits of thiol-ene photopolymerization [13]. Rapid step-growth thiol-ene photopolymerization reactions are insensitive to oxygen and offer low shrinkage [14]. Thiol-ene photopolymerization is a promising alternative to acrylate-based polymerization reactions [15]. It offers linear and cross-linking polymerizations depending on the number of functional groups of thiols, alkenes, presence of crosslinker, etc. Various thiols can be used in combination with different alkene compounds, creating multifunctional structures of polymers. Thiol-ene reactions also offer a synthesis of biocompatible and biodegradable polymers [16,17].

Polymer nanoparticles produced by aerosol thiol-ene photopolymerization possess accessible -SH groups, according to findings from previous work [18]. These functional groups can be an excellent tool for thiol-ene “click” reactions [19,20]. Cancer targeting molecules, biomarkers, antibodies, etc. can easily be conjugated onto the surface of nanoparticles via these available groups. Maleimides present a promising tool for bioconjugation, especially with thiol-groups of nanoparticles through Michael-type addition [21]. A great variety of commercially available derivatives of maleimide (e.g., biotin-maleimide) can be applied for further functionalization. Conjugation of biotin-maleimide to the silver-polymer nanoparticle hybrids allows conjugation of streptavidin through high-affinity biotin-streptavidin binding [22].

Cancer nanotechnology is a fast-developing field mainly focused on the use of nanomaterials in diagnostics and treatment of cancer by addressing selectivity—one of the major issues in present cancer treatment techniques [23,24,25,26]. The tunable properties of nanomaterials are the key to their versatile applications in targeting cancer. Despite their small size (1–1000 nm), nanoparticles can carry a great quantity and variety of drug and antibody molecules; nanoparticles can be functionalized to specifically target cancer cells, and are inert to resistance mechanisms of the organism [27].

Strong optical absorbance and scattering properties of noble metal nanoparticles make them a desirable tool for combination therapies [28,29,30]. A great amount of research has been carried out to establish applications of gold nanoparticles for in vitro assays, in vitro and in vivo imaging, cancer therapy, and drug delivery [31]. Localized surface plasmon resonance (LSPR) is an effect observed in metal nanoparticles, resulting in radiation scattering and absorption of light [32]. LSPR and other optical properties of metal nanoparticles usually depend on the size and shape of nanoparticles and their aggregation [33,34]. Localized heating that occurs due to LSPR carries on to apoptosis of surrounding cells and/or releases therapeutics. LSPR of gold nanoparticles majorly contributes to the use of photoimaging and photothermal therapy of cancer tumors [35]. LSPR wavelengths of gold and silver nanoparticles are in the range of the visible spectrum of light [32]. Although, silver and gold nanoparticles have similar plasmon resonance sensitivity, sharper and more intense resonance peaks of silver nanoparticles are advantageous in sensor applications [36]. Silver nanomaterials have already been applied in cancer research due to their bactericide antimicrobial features [28]. However, their cytotoxicity and genotoxicity due to the presence of both silver ions (Ag^+^) and reactive oxygen species (ROS), generated by silver nanoparticles incorporated in human cells [37,38], have restricted their wide application in cancer treatment. The optical properties of silver nanoparticles in cancer diagnostics have only been used in a limited number of studies [39,40]. Optical properties of silver nanoparticles are better than those of gold nanoparticles [41], therefore the use of silver nanoparticles in cancer diagnostics might result in higher sensitivity compared to gold nanoparticles. Less research was found on the sensor applications of silver nanoparticles for cancer diagnostics and treatment, compared to gold nanoparticles.

In our recent paper [18], we investigated several thiol-ene monomer combinations that worked well in combination with aerosol photopolymerization to form spherical individual nanoparticles with reactive -SH groups [18]. In the present study, we aim to establish a working combination to produce spherical individual polymer nanoparticles that contain individual silver nanospheres inside. The resulting particles should possess accessible -SH groups for conjugation of biomolecules, and silver nanoparticles provide, e.g., plasmon resonance properties for imaging techniques.

## 2. Materials and Methods

### 2.1. Chemicals

Trimethylolpropane tris (3-mercaptopropionate) (Trithiol, 95%, Sigma-Aldrich, St. Louis, MO, USA) and trimethylpropane triacrylate (TMPTA, Sigma-Aldrich, contains 600 ppm monomethyl ether hydroquinone as inhibitor) were used as thiol and alkene monomers, respectively. 2-Methyl-4′-(methylthio)-2 morpholinopropiophenone (MT-2MP, 98%, Sigma-Aldrich, St. Louis, MO, USA) was used as a photoinitiator. Silver nanoparticles of different grades were used in preparation of the spray solution: silver nanoparticles ink (AgINK, 115 nm (d_90_), 70 nm(d_50_), Merck KGaA, Darmstadt, Germany, 50 wt. % dispersion in tripropylene glycol mono methyl ether), dried silver nanoparticles (Ag25 nm, Nanocomposix, Econix PVP coated), dried silver nanoparticles (Ag50 nm, Nanocomposix, Econix PVP coated). Stabilization of silver nanoparticles was carried out using the following compounds: ± α-Lipoic acid (α-LA, ≥98%, Sigma-Aldrich, St. Louis, MO, USA), 11-Mercapto-1-undecanol (MUL, 99%, Sigma-Aldrich, St. Louis, MO, USA). 5,5′-dithiobis-(2-nitrobenzoic acid) (DTNB, ReagentPlus^®^, 99%, Merck KGaA, Darmstadt, Germany) was among the main components for Ellman’s reaction for the determination of -SH groups. Gold nanoparticles (Au@Mal, maleimide functionalized, conjugation kit, CytoDiagnostics Inc., Sigma-Aldrich, St. Louis, MO, USA), Bio-ready Gold Nanospheres-Streptavidin (Au@Strep, Nanocomposix, San Diego, CA, USA), Atto 425-Streptavidin (425-St, λ_em_ = 477 nm, Merck KGaA, Darmstadt, Germany), Biotin-Maleimide (b-M, ≥95%(TLC) powder, Merck KGaA, Darmstadt, Germany), iFluor-Maleimide-350 (i-FM, λ_em_ = 441 nm, AAT Bioquest Inc., Sunnyvale, CA, USA) were used for (bio)conjugation reactions. Phosphate buffered saline (PBS) pH 7.4 was used as a buffer for streptavidin conjugation (137 mM Sodium Chloride, 2.7 mM of Potassium Chloride, 10 mM of Di-sodium hydrogen phosphate, and 1.8 mM of Potassium di-hydrogen phosphate in distilled water). Dimethyl Sulfoxide (DMSO, anhydrous, ≥99.7%, Carl Roth GmbH, Karlsruhe, Germany) was used as a solvent.

### 2.2. Aerosol Photopolymerization

The schematic representation of the aerosol photopolymerization setup is presented in Figure 1. The spray solution placed inside the aerosol generation unit (ATM 220, Topas GmbH, Dresden, Germany) is atomized under the stream of pressurized nitrogen, forming a droplet aerosol, which, subsequently, passes to the photoreactor unit. In the photoreactor, droplets polymerize under the irradiation of UV-fluorescence lamps (λ_max_ = 312 nm, T-15.C, Vilber Lourmat, incident irradiance E = 15.4 mW/cm^2^ (measured via UV-Pad-E from Opsytec Dr. Gröbel GmbH)) to form polymer nanoparticles, which are collected on 0.1 µm pore size Durapore^®^ hydrophilic membrane filter (Millipore, Mersck KGaA, Darmstadt Germany) [18]. During the process, while the volatile solvent in the spray solution is partially evaporated, every 30 min after the start of the reaction, the solvent is added to compensate for the evaporated amount.

### 2.3. Experimental Methods

#### 2.3.1. Preparation of Spray Solutions

The spray solutions prepared for the aerosol photopolymerization contain thiol-ene monomers, organic volatile solvent, photoinitiator, silver nanoparticles, and if required, a silver nanoparticle stabilizer. The formulations of the spray solutions used within the scope of this paper are presented in Table 1. For the spray solution preparation, silver nanoparticles were first introduced into the solvent (with stabilizer), then either stirred or sonicated (according to stabilizer of choice) until a colloidally stable dispersion was obtained. Then, the monomers (0.7108 g of Trithiol and 0.9559 g of TMPTA) were introduced into the dispersion, followed by the addition of the photoinitiator (0.0017 g) right before the photopolymerization process.

Spray solutions were treated differently depending on chosen stabilization techniques. In the case of α-LA, the silver nanoparticles were stirred with a 0.1 mM α-LA ethanol solution for 30 min. The same stabilization method was used with MUL. Stabilization via sonication was carried out with the spray solution flask (100 mL) immersed in an ultrasonic bath (Sonorex Digital 10P, 80% amplitude, 3 min, 22 °C) during each solvent addition step.

#### 2.3.2. Biofunctionalization

The nanoparticles obtained via aerosol photopolymerization were tested for the presence of reactive -SH groups using Ellman’s reaction [18]. The solution of DTNB in phosphate-buffered saline (PBS) was prepared and added to the dispersion of nanoparticles. The instant reaction that showed the appearance of yellow color confirms the presence of unreacted -SH groups. Poly(TMPTA) nanoparticles (without -SH groups) were used as a blank experiment.

Maleimide conjugated gold nanoparticles (Au@Mal) that can be visualized via electron microscopy were used to investigate the availability of binding sites on the surface of the polymer nanoparticle layer (Figure 2A(1)). To avoid misinterpretation of electron microscopy images, the plain polymer nanoparticles without AgNPs were used (formula NC-0, Table 1). A total of 1 mg of Au@Mal was dissolved in 1 mL of reagent buffer (provided with the conjugation kit) and added to 5 mg in 1 mL of dimethyl sulfoxide (DMSO, anhydrous). The reaction was carried out for 12 h, followed by purification via centrifugation. The particles were quenched with 3 × 5 mL of EtOH.

The availability of binding sites was also measured for the reaction with streptavidin, by consecutive addition of biotin-maleimide (b-M) and streptavidin conjugated to gold nanoparticles (Au@Strep) (Figure 2A(2)). A total of 1 mg of b-M was dissolved in 1 mL of DMSO and added dropwise to the dispersion of 5 mg of NC-0 NPs in 1 mL of DMSO. The reaction was carried out for 12 h at room temperature. The particles were purified with 3 × 5 mL of EtOH and left to dry. The dry b-M-NC-0 nanoparticles were then dispersed in 1 mL of 3% DMSO in phosphate-buffered saline (PBS) (pH 7.4) and introduced into the reaction with 200 µL of Au@Strep dispersion. The reaction was left stirring for 12 h and then purified via centrifugation with 3 × 5 mL of PBS (pH 7.4).

The reaction of maleimide-fluorophore (i-FM) with nanoparticles (Figure 2B(1)) was also carried out in DMSO. At first, 1 mg of i-FM was dissolved in 1 mL of DMSO. Then, 10 mg of NC-7 nanoparticles were dispersed in 4 mL of DMSO and stirred. The solution of i-FM was added to the dispersion of nanoparticles dropwise under vigorous stirring. The mixture was left to react overnight. Afterwards, the reaction mixture was purified by centrifugation with 2 × 10 mL of DMSO and 2 × 10 mL of EtOH. Each supernatant was tested via fluorescence spectroscopy for the presence of unreacted fluorescein-maleimide.

The reaction of nanoparticles with biotin-maleimide was carried out in the same way as the abovementioned reaction with NC-0. The resulted (b-M)-NC-7 nanoparticles were used in further reactions with fluorescein-streptavidin (Figure 2B(2)).

The reaction of (b-M)-NC-7 nanoparticles with 425-St was carried out in PBS (pH 7.4). A total of 1 mg of 425-St was dissolved in 1 mL of PBS. A total of 10 mg of nanoparticles were dispersed in PBS stirred for 10 min and sonicated in an ultrasonic bath (Sonorex Digital (Bandelin Electronic GmbH & Co. KG, Berlin, Germany) 10P, 100% amplitude, 5 min, 22 °C). The 425-St was added dropwise to the nanoparticle dispersion under vigorous stirring and stirred overnight. The reaction mixture was purified by centrifugation with 2 × 20 mL of PBS and then 2 × 20 mL of EtOH-PBS mixture.

### 2.4. Analytical Methods

#### 2.4.1. Scanning Electron Microscopy

SEM was used to observe the size, shape, degree of aggregation, and encapsulation of AgNPs inside the polymer matrix of Ag@poly(Trithiol-TMPTA) nanoparticle hybrids produced via aerosol thiol-ene photopolymerization. The samples for the analysis were prepared as follows: the dry nanoparticle product was dispersed in acetone and stirred for 30 min, 100 µL of the dispersion was distributed on silicon wafers, dried, and sputtered with platinum. A LEO1530 (Carl Zeiss Microscopy GmbH, Jena, Germany). SEM was used in all experiments and images were taken at magnifications of 2000, 10,000, and 25,000. The images were recorded at working distance 5.5–6 mm, the acceleration voltage of the microscope was 5 keV.

#### 2.4.2. Transmission Electron Microscopy

TEM was used to observe the degree of aggregation and location of silver nanoparticles within the polymer matrix in the obtained Ag@poly(Trithiol-TMPTA) nanoparticles and conjugation of bioconjugated gold nanoparticles onto the surface of poly(Trithiol-TMPTA) nanoparticles. The samples for the analysis were prepared as follows: an ultrathin 3 nm carbon film on Lacey carbon film 300 copper mesh TEM grid was placed on top of the membrane filter inside the filter housing during the polymerization reaction for five minutes, collecting the nanoparticles. The TEM grids were analyzed with a CM 200 FEG (Phillips, Amsterdam, Netherlands) microscope (operated on an acceleration voltage of 200 keV). The images were recorded by zooming into individual nanoparticles or nanoparticle clusters.

#### 2.4.3. UV-Vis Spectrophotometry

The quantitative analysis in Ellman’s test was carried out in 96 well UV-Star^®^ microplates (Greiner Bio-One) using a Tecan Infinite^®^ UV-VIS Spectrophotometer (Tecan Group Ltd., Männedorf, Switzerland). Samples were prepared as follows: 5 mg of nanoparticles were dispersed in 1 mL of PBS pH 7.4; 100 µL of this dispersion was placed inside a well and 100 µL of DTNB dissolved in PBS pH 7.4 was added to the same well. The absorption measurements were carried out at 412 nm wavelength at 22 °C.

#### 2.4.4. Fluorescence Spectroscopy (FS)

Fluorescent properties of fluorescein-maleimide-labelled nanocomposites were analyzed using an OceanOptics Maya2000 Pro (Ocean Insight, Orlando, FL, USA) fiber optic spectrophotometer with a quasi-monochromatic LED, λmax = 365 nm (HAMAMATSU, LC1) and a long-pass filter (cut on 400 nm). Samples were prepared by placing 2.5 mL of the appropriate solution/dispersion into 10 mm Hellma™ Suprasil™ cuvettes.

#### 2.4.5. Fluorescence Microscopy (FM)

Fluorescein-maleimide and Fluorescein-Streptavidin labelled nanocomposites were visualized using Olympus IX81F-ZDC2 Confocal Laser Scanning Microscope (Olympus Corporation, Shinjuku, Tokyo, Japan) at 405 and 488 nm, respectively. A 525/50 Filter was used during the analysis. Sample preparation for fluorescence microscopy included dispersion of labelled nanocomposites in appropriate solvents (EtOH or PBS), placing 50 µL of this dispersion onto the 76 × 26 mm (thickness 1 mm) microscope slides covered with 18 × 18 mm cover glass.

## 3. Results and Discussion

### 3.1. Silver@poly(Thio-Ether) Nanoparticle Hybrids

Uncoated silver nanoparticles (AgNPs) are prone to aggregate in dispersed media in the presence of alkenes and other compounds [42], depending on pH, ionic strength, and electrolyte composition of the dispersion [43]. Therefore, stabilization techniques are required to prevent their aggregation. Among several stabilization mechanisms available, steric and electrostatic stabilization methods have shown to be more effective in preventing the aggregation of silver nanoparticles [44]. In the present study, stabilization with ±α-lipoic acid [45] and steric stabilization with 11-mercapto-1-undecanol [46] were examined.

Various grades of AgNPs were considered in order to establish a convenient method of synthesis. Ink dispersion of silver nanoparticles in tripropylene glycol monomethyl ether (AgINK) was one of the most available and cost-effective variants. Polyvinylpyrrolidone (PVP)-coated dried silver nanospheres (Ag25 nm and Ag50 nm) although less cost-effective, provided stabilized silver nanoparticles less prone to aggregation. A total of 70–90% of the weight of Ag25 nm and Ag50 nm was PVP and only 10–30% was silver. Ag25 nm and Ag50 nm also possessed narrow particle size distributions, whereas AgINK revealed a broad size distribution and various shapes of nanoparticles (small triangles, ovals, squares). Therefore, the stabilization of AgINK was the primary goal.

The first step in every experiment was to stabilize the dispersion of monomers and AgINK. Several experiments showed aggregation of nanoparticles into visible flakes during the first 30 min of the atomization process. The attempts to stabilize the nanoparticles in dispersion included use of different stabilizers, dialysis with other solvents, as well as intermittent sonication. Nanoparticle hybrids obtained from NC-1 (no silver stabilization) formulation have shown aggregation of silver nanoparticles employing TEM (see Figure 3). In addition, silver nanoparticles appeared on the SEM (and the TEM) images on the surface of nanoparticles—they are not completely encapsulated inside the polymer matrix, which might be an issue in biomedical applications. The formation of flakes (big aggregates of silver nanoparticles) in the spray solution during the photopolymerization process can be associated with the observation of silver nanoparticle aggregates on the surface as well as within the polymer (image not presented).

The aggregation behavior of silver nanoparticles is well studied [42,44]. According to research by Guzman-Soto et al. [47], α-LA can improve the oxidative stability of silver nanoparticles. AgINK stirred in combination with EtOH and α-LA for 30 min prior to the atomization process showed better stability of the spray solution throughout the reaction. The nanohybrids obtained from this photopolymerization process showed a lower tendency to aggregation of AgNPs (Figure 3, NC-2). The addition of α-LA to the spray solution showed minimal change in the appearance of polymer nanoparticles on the SEM, although it was discovered during this research that there is a certain concentration of α-LA beyond which the polymer nanoparticles started to agglomerate. Therefore, the concentration of α-LA was kept in a range of 10–15 mL of 10 mM solution (in EtOH). Stabilization with α-LA resulted in individual AgINK nanoparticles fully encapsulated inside the polymer particles. Smaller quantities of silver nanoparticles were observed inside polymer nanoparticles on TEM images.

Other methods to stabilize silver nanoparticles were considered as well. 11-Mercapto-1-undecanol (MUL) was used as a steric stabilizer for silver nanoparticles in ink [48]. MUL showed substantial improvement of the dispersion of silver nanoparticles within the polymer matrix. In comparison to α-LA, silver nanoparticles with MUL show a lower tendency for aggregation. This can be explained by the formation of a MUL layer around the silver nanoparticles [48]. TEM images reveal that the silver nanoparticles are both individual and fully encapsulated inside the polymer matrix (Figure 3, NC-3). In addition, silver nanoparticles were not present on the surface of polymer nanoparticles in SEM images.

Stabilization via sonication of silver nanoparticle dispersions attempted to break down the aggregates of silver formed during the aerosol generation process. Ultrasonic agitation of the spray solution resulted in the temporary breaking of aggregates of silver nanoparticles, but had to be repeated every 30 min as aggregates redeveloped. SEM and TEM images (Figure 3, NC-4) of nanocomposites from sonicated spray solutions show reduced aggregation of silver nanoparticles compared to non-sonicated spray solutions (NC-1).

The influence of the solvent on the formation of polymer nanoparticles and aggregation of silver nanoparticles was studied by replacing ethanol with 1-propanol. SEM images of the hybrid nanoparticles particles (Figure 3, NC-8) obtained from the spray solution with 1-propanol showed increased agglomeration of the polymer nanoparticles, whereas the silver nanoparticles exhibited lowered aggregation and full encapsulation within the polymer matrix. Some silver nanoparticles still appeared outside the polymer nanoparticles. The TEM image of lower magnification (with several polymer nanoparticle hybrids is presented in Appendix A. Only one image of NC-8 polymer nanoparticle hybrids is shown as an example of how the hybrids are observed on a TEM image from a lower magnification, proving that the majority of polymer hybrid had an average amount of silver nanoparticles inside (3–4 AgNPs per hybrid). However, some smaller polymer particles appear to have no AgNPs.

PVP-coated silver nanoparticles are of higher quality than AgINK nanodispersion; Ag25 and Ag50 NPs did not require pre-dispersion due to the PVP coating that prevented aggregation of silver nanoparticles in the spray solution. Nevertheless, the PVP coating makes up for 70–90% of Ag25 nm and Ag50 nm nanoparticles, which might be a downside for potential applications. Ag25 nm and Ag50 nm behaved differently when incorporated in the polymer matrix. Samples with Ag25 nm (Figure 3, NC-5) showed high amounts of aggregates of silver nanoparticles, but a lower aggregation of silver NPs was observed when α-LA was added (Figure 3, NC-6), whereas Ag50 nm (Figure 3, NC-7) samples revealed desired individual and well-distributed silver NPs within the polymer matrix.

The size distributions of silver nanoparticles, gold nanoparticles and polymer nanoparticle hybrids are presented in Appendix A. In Appendix A, a histogram of NC-7 polymer nanoparticle hybrids is presented, where d_50_ of the particles is 325 nm. The size of the nanoparticle hybrids can always be narrowed by varying the parameters of the aerosol photopolymerization, e.g., solvent amount. As presented in our previous work [18], increasing solvent amount of the spray solution the size of polymer nanoparticles can be narrowed to fit appropriate applications.

Appendix A shows the size distributions of AgINK, Ag50 nm, and Ag25 nm nanoparticles, to show the advantages of using Ag50 nm for further applications due to the narrower size distribution, compared to AgINK, where the size distribution is very broad. However, more experiments were carried out with cost-effective AgINK nanoparticles compared to more expensive and purer Ag50 nm.

Encapsulation of silver nanoparticles inside the polymer matrix is a possible strategy of stabilization, eliminating their cytotoxicity in future applications [49].

### 3.2. Nanoparticle Binding Sites

To visualize the presence of the accessible binding sites on the surface of polymer nanoparticles employing TEM, a one-step conjugation to gold NPs-maleimide (Au@Mal) and a two-step conjugation to biotin-maleimide and gold NPs-streptavidin (Au@Strep) were carried out with NC-0 polymer nanoparticles. The conjugation to Au@Mal resulted in the formation of polymer nanoparticles with many gold nanoparticles located on the surface of the polymer (Figure 4 left). This evinces the presence of reactive -SH groups. The two-step conjugation (1. biotin-maleimide, 2. streptavidin-gold) leads to the conclusion that a sufficient number of binding sites on the polymer nanospheres is available (Figure 4 right). From the presented results, the number of binding sites in one-step conjugation seems higher than in a two-step conjugation process. This might be a result of higher steric accessibility of Au@Mal to the -SH groups, compared to the bulkier streptavidin in Au@Strep. Sizes of Au@Mal and Au@Strep nanoparticles were evaluated via image analysis of TEM micrographs and the size distributions of these nanoparticles are presented in Appendix A.

### 3.3. Bioconjugation of Silver@poly(Thio-Ether) Nanoparticles

The presence of reactive -SH was tested via Ellman’s reaction on NC-7 polymer nanoparticle hybrids. Solution of DTNB mixed with the dispersion of NC-7 results in distinctive yellow coloration. To ensure that the coloration is not due to unreacted thiol monomers within the polymer matrix, a purification procedure was carried out in acetone (centrifugation, 10,000× *g*, 3 × 20 mL). Purified and non-purified nanoparticles show different results with DTNB. In Table 2, absorption of DTNB and nanoparticles (reacted with DTNB) at 412 nm are shown.

Absorption of 0.7 mM DTNB (the reagent solution) is presented for comparison. The concentration of -SH groups is calculated using the Beer-Lambert law, taking 14,500 M^−1^ cm^−1^ as the extinction coefficient for DTNB in buffer solutions of pH = 7.4. The stray light correction and absorption contrast were carried out with poly(TMPTA) nanoparticles. Purification of the nanoparticles results in an about ten-fold decrease of determined -SH groups within the polymer matrix of the nanoparticles. This decrease can be due to oxidation of -SH groups during purification procedure and/or removal of unreacted thiol monomers from the nanoparticles during purification.

Silver-polymer nanocomposites (NC-7) were conjugated with fluorescein labelled maleimide (i-FM), and with biotin-maleimide (b-M) and fluorescein-labelled streptavidin (425-St), sequentially. Functionalization with maleimide-fluorophore was considered to confirm the presence of -SH groups on the nanoparticles, whereas functionalization with streptavidin-fluorophore confirms the possibility of a multi-step conjugation. Functionalized maleimide reacts with thiol groups of nanoparticles via thiol-Michael addition in a one-step (bio)conjugation. The basis of the first step of the two-step bioconjugation is also thiol-ene “click” chemistry (biotinylation), whereas the second step is biotin-streptavidin binding (Figure 2). The successful formation of bioconjugated particles was visualized by Fluorescence Spectrometry (FS) and Fluorescence Microscopy (FM).

Purified fluorescence-labelled nanoparticle hybrids (NC-7-i-FM) show intense fluorescence with a maximum at ~430 nm via FS (Figure 5 left). The fluorescence spectrum of pure i-FM in DMSO shows a narrow peak at 434 nm. This peak can also be well observed for supernatants of the first four purification steps. The apparent stray light peak of NC-7-i-FM is due to the presence of nanoparticles in purified dispersion and can be observed in Figure 5 (λ_max_ ≈ 385 nm), in contrast to the supernatant spectrum. FM images of these nanoparticles at 405 nm excitation wavelength also showed the presence of nanoparticles tagged with fluorescence label (Figure 5 right). No fluorescence was observed for non-conjugated nanoparticle hybrids (NC-7, not presented).

The degree of conjugation of b-M to the nanoparticles is unknown, but the results of the subsequent reaction with 425-St evince presence of fluorescence-labelled material (see Figure 6). The nanoparticles at 405 nm excitation wavelength are not as well dispersed as for NC-7-i-FM. In the reaction of NC-7-b-M with 425-St, a PBS solution was used instead of DMSO to control the pH and ionic strength of the experiment, following the recommendations of the supplier of 425-St. The NC-series nanoparticle composites dispersed in polar solvents such as PBS form flakes, in contrast to a stable dispersion of these nanocomposites in DMSO. Therefore, the microscopy analysis did not show well-dispersed nanoparticles. Fluorescence spectra show an apparent stray light peak of NPs at ~385 nm and a fluorescence peak at ~475 nm in the purified dispersion of (NC-7-b-M-St) (Figure 6) streptavidin-conjugated nanoparticles, which confirms successful two-step bioconjugation.

The above-presented fluorescence spectroscopy and microscopy results evince the availability and accessibility of functional -SH groups on the surfaces of Ag@poly(Trithiol-TMPTA) nanoparticle hybrids and their subsequent biofunctionalization. We anticipate that the cytotoxicity of silver nanoparticles is reduced due to the availability of -SH groups within the polymer matrix [50]. They have a high adsorption capacity of Ag^+^, subsequently stabilizing the AgNPs [51] and suppressing their cytotoxicity. However, the research into the cytotoxicity of the silver nanospheres within the polymer particles will be a topic of future work. The cytotoxicity can be evaluated by introducing the produced hybrid particles to living cells and performing cell viability assays. These results can be compared to the viability of cells with pure silver nanospheres. The reactive SH-groups are an otherwise effective tool for the functionalization of complex biomolecules, e.g., to target cancer cells specifically. The size of the produced nanocomposites can be toggled [18] to match the specific applications.

## 4. Conclusions

Hybrid nanomaterials are systems that combine unique physical and chemical properties of their single components and thus can be used for various applications. It is necessary to look for convenient, sustainable methods of synthesis, control physical properties, and investigate the range of new possible applications of the hybrid nanomaterials.

Aerosol thiol-ene photopolymerization provides an easy, non-toxic, eco-efficient method for the synthesis of spherical polymer matrix nanocomposites with silver nanoparticles inside. As with many metal nanoparticles, stabilization of silver nanoparticles was required to avoid their aggregation. All stabilization strategies studied in this paper proved to be effective in forming individual silver nanoparticles well dispersed within the polymer matrix. The application of α-Lipoic acid provided slightly better results, compared to 11-mercapto-1-undecanol and stepwise sonication of cost-effective AgINK. Polyvinylpyrrolidone-coated silver nanoparticles (Ag25 nm) should be stabilized with α-LA for the synthesis of nanocomposites with well-dispersed silver nanoparticles. Ag50 nm did not require any additional stabilizer. Ag50 nm showed exceptional compatibility with aerosol thiol-ene photopolymerization for the synthesis of hybrid nanoparticles.

The obtained nanocomposites possess -SH groups on their surface and can, thus, be used in bioconjugation. The reactivity of binding sites was successfully confirmed by using gold-tagged and fluorescence-labelled biomolecules.

Nanoparticle hybrids produced in this study can find applications in cancer diagnostics and treatment, as well as many other areas of biomedicine and biosensors.

## Figures and Tables

**Figure 1 nanomaterials-12-00577-f001:**
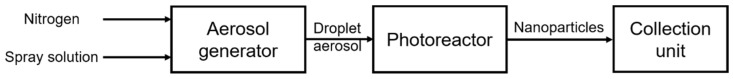
Schematic representation of the aerosol photopolymerization setup.

**Figure 2 nanomaterials-12-00577-f002:**
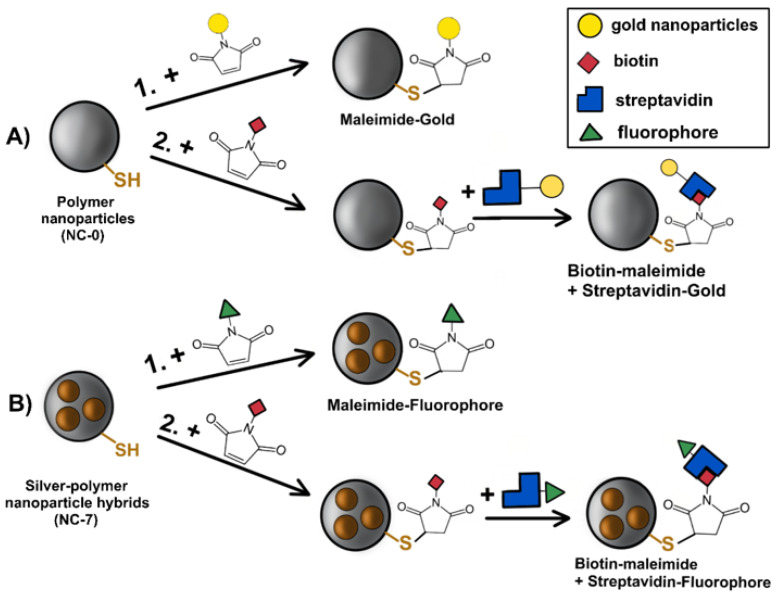
Diagram of bioconjugation of gold nanoparticles onto the surface of polymer nanoparticles (**A**) and bioconjugation of fluorescence-tagged molecules to the hybrid nanoparticles (**B**). Legend is in the top right corner.

**Figure 3 nanomaterials-12-00577-f003:**
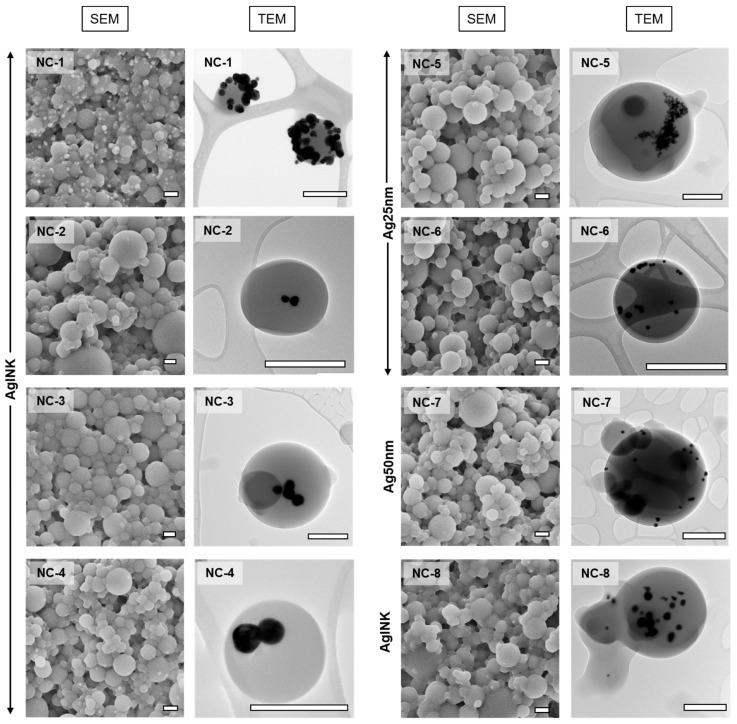
SEM and TEM images of the Ag@poly(Trithiol-TMPTA) nanoparticles produced via Aerosol Photopolymerization. The nomenclature corresponds to spray solution formulations presented in Table 1. Black particles on TEM represent silver nanoparticles, the grey spheres represent polymer layer. The small radiant white spots on SEM represent the silver nanoparticles, whereas bigger grey spheres are polymer matrix. The scalebars correspond to 0.5 µm. Vertical labels on the left correspond to the type of silver nanoparticles used in the reaction and agrees with the data in Table 1.

**Figure 4 nanomaterials-12-00577-f004:**
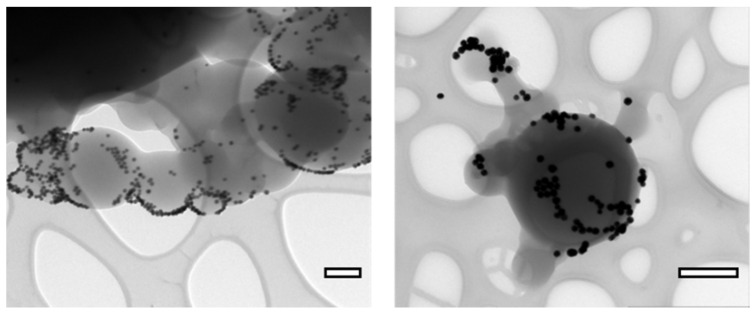
Maleimide-gold NPs conjugated to poly(Trithiol-TMPTA) nanoparticles (**left**) and Streptavidin-gold NPs conjugated to bM-poly(Trithiol-TMPTA) nanoparticles (**right**) as an evidence of presence of reactive -SH groups on the surface of these nanoparticles (for Au-Mal@poly(Trithiol-TMPTA)) and effective biotin-Streptavidin bonding (for Au-Strep@bM-poly(Trithiol-TMPTA)). The scalebars correspond to 0.5 µm.

**Figure 5 nanomaterials-12-00577-f005:**
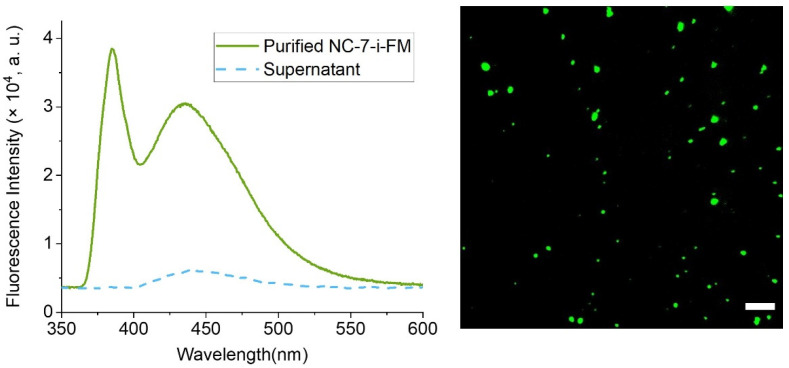
Fluorescence Spectra (**left**) and Fluorescence Microscopy image (**right**) of i-Fluor-Maleimide conjugated NC-7 nanoparticles (NC-7-i-FM). The solid green line represents the spectrum of purified NC-7-i-FM NPs dispersed in ethanol (1 mg/5 mL) and the dashed blue line represents the spectrum of the last supernatant of these nanoparticles after the purification. Fluorescence microscopy image (λ_em_ = 405 nm, scalebar 10 µm) confirms that the NC-7-i-FM nanoparticles are the cause of fluorescence and sporadic agglomerates of the nanoparticles can be observed.

**Figure 6 nanomaterials-12-00577-f006:**
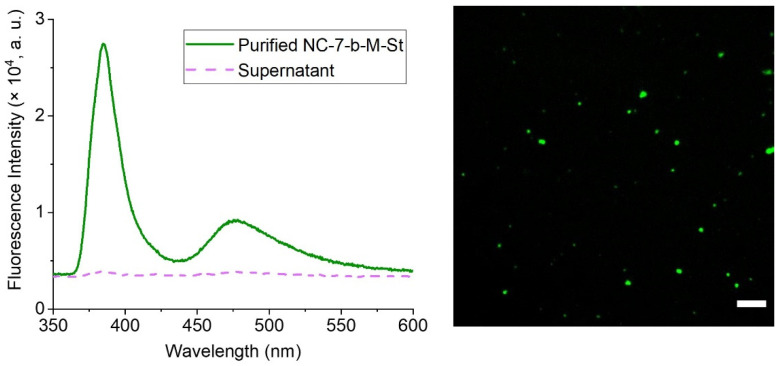
Fluorescence Spectra (**left**) and Fluorescence Microscopy image (**right**) of biotin-maleimide and Atto-425-Streptavidin conjugated NC-7 (NC-7-b-M-St) nanoparticles. The solid green line represents the spectrum of purified NC-7-b-M-St NPs dispersed in 3% DMSO in 100 mM phosphate buffer (pH 7) (1 mg/5 mL) and the dashed line represents the spectrum of the last supernatant of these nanoparticles after the purification. Fluorescence microscopy image (λ_em_ = 450 nm, scalebar 10 µm) confirms that the NC-7-b-M-St nanoparticles are the cause of fluorescence and sporadic agglomerates of the nanoparticles can be observed.

**Table 1 nanomaterials-12-00577-t001:** Formulations of spray solutions used to produce silver poly(thio-ether) nanoparticle hybrids. Amounts of monomers and photo-initiator were kept the same in all presented spray solution formulations.

Spray Solution	Silver NPs	Silver NPs Quantity	Stabilization Method	Stabilizer Concentration	Solvent-Quantity (g)
NC-0	-	-	-	-	EtOH—50
NC-1	AgINK	0.17 g	-	-	EtOH—50
NC-2	AgINK	0.17 g	α-LA	0.1 g	EtOH—50
NC-3	AgINK	0.017 g	MUL	0.01 g	EtOH—50
NC-4	AgINK	0.17 g	Sonication	-	EtOH—50
NC-5	Ag25 nm	0.05 g	-	-	EtOH—50
NC-6	Ag25 nm	0.1 g	α-LA	0.08 g	EtOH—25
NC-7	Ag50 nm	0.01 g	-	-	EtOH—50
NC-8	AgINK	0.17 g	α-LA	0.13 g	n-PrOH—50

**Table 2 nanomaterials-12-00577-t002:** Absorption and approximate concentration of free -SH groups in purified and non-purified nanoparticles.

Sample	Absorption at 412 nm	Concentration of -SH Groups (M/mg)
DTNB (0.7 mM in PBS)	0.2056	-
Purified NPs (1 mg/mL in PBS)	0.2793	0.009 × 10^−3^
Non-purified NPs (1 mg/mL in PBS)	0.8896	0.087 × 10^−3^

## Data Availability

The data presented in this study are available on request from the corresponding author.

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
