# Peer review of "Synthesis of Spherical Nanoparticle Hybrids via Aerosol Thiol-Ene Photopolymerization and Their Bioconjugation"

_nanomaterials, 2022, doi:10.3390/nano12030577_

Round 1

Reviewer 1 Report

It is a scientific consensus that silver and gold nanoparticles can be potent
bactericidal, fungicidal, and even virucidal agents. The aggregation of these
nanoparticles is a problem that hinders the antimicrobial mechanism,
thus decreasing the activity. The question is what is the property of silver
or gold nanoparticles developed in these conjugation systems?
if according to your text:

...stabilizing the AgNPs [48] and suppressing their cytotoxicity. The reactive SH-groups are an otherwise effective tool for the functionalization of complex biomolecules e.g., to target cancer cells specifically. The size of the produced nanocomposites can be toggled [18] to match the specific applications.

... a synthesis of spherical polymer matrix nanocomposites with silver nanoparticles inside. As with many metal nanoparticles, stabilization of silver nanoparticles was required to avoid their aggregation. (OK)

But so a large-cap would avoid any nanoparticle reactivity, activity, or mechanism for Ag+ release.

Author Response

Dear Reviewer,

Thank you for time and effort to review our mauscript. We have addressed all concerns you have about the context of our paper and applied changes to the text according to your and other reviewer's comments.

  • Silver nanoparticles were explored for their antimicrobial properties in other research papers.
  • The aim of this particular work was to encapsulate the silver nanoparticles within the functional polymer matrix. Silver nanoparticles have substantial optical properties (LSPR) that can be used in cancer diagnostics and treatment, but for that purpose, the cytotoxicity of silver nanoparticles has to suppressed. We attempt to entrap the silver nanoparticles within the polymer matrix which can also serve as a tool for functionalization.
  • Indeed, we did not look into how the cytotoxicity was suppressed after polymerization, but this would be a purpose of future works. (see lines 396-402).

In the attached document you can find the manuscript with added/modified text in a form of a PDF file.

Kind regards,

Authors

Reviewer 2 Report

The authors present a methodology whereby silver nanoparticles are dispersed within a photo-polymerizable matrix, which is then subject to aerosol photopolymerization. The resulting nanocomposite particles can then be grafted with biologically relevant molecules, thus facilitating transport and tracking (via silver's optical properties).

The work is generally well presented and described, though I have some reservations I invite the authors to address.

General comments:

  • The authors point out that nanoparticle-polymer hybrid materials have been previously obtained (by their group and others) via aerosol photopolymerization. What, therefore, is the novelty being presented herein? The authors are advised to clarify the challenged inherent to using silver particles early in the introduction, to better justify the work.
  • Have the authors tested if their final particles release Ag+ (cytotoxic) when suspended in a biologically relevant medium?

  • Many stabilization molecules are considered on Ag particles, but only studied after photopolymerization. Have the authors considered studying colloidal stability using DLS or other well-established techniques prior to polymerization? This would allow them to identify if it is the stabilizer or the photo-processing that leads to aggregation.

Specific comments:

  • Line 55 - the statement "Polymer nanoparticle hybrids produced by aerosol photopolymerization possess ac-55 cessible -SH groups [18] " is an over-generalization. That particular case presented -SH groups, but this is not a definite result for all types of aerosol photopolymerization
  • Line 78 - It is pertinent to note that the optical properties of noble metal nanoparticles are size-dependent. In other words, fine control on particle size and aggregation is necessary

  • Line 129 - the authors must report the irradiance of the UV lamp, at the distance of interest, to allow reproduction of their work

  • Line 141 - "stirred or sonicated" are two vastly different approaches to dispersion. Proper dispersion is a key factor to ensure that the authors attain their objective (optically active Ag nanoparticles in a polymer matrix). More information is given in lines 146-149, but this remains incomplete. Container volume should be given, as well as sonication energy when used (see for example the guidelines provided by Girard et al. in Ultrasonics Sonochemistry 71, 105378). Similar comment on lines 188-189

  • Section 2.4 - Accelerating voltage, current and working distance should be reported for electron microscopy

  • Line 269-271 - this statement is not clear to the reviewer. Moreover, the term "superb stabilization" is non-scientific and qualitative.

  • Lines 283-284 - This points to an unstable suspension. In all likelyhood, insufficient stabilizer is present.

  • Lines 391-395 - This is a blanket, unvalidated statement. Please refer to my previous general comment pertaining to cytotoxicity

Author Response

Dear Reviewer,

Thank you for taking time and putting effort to review our mauscript. We have addressed all concerns you have about the context of our paper and applied changes to the text according to your comments (the modified text can be found in the attached PDF file in form of the highlighted text). Below are replies to your comments.

  1. polymer nanoparticle hybrids synthesized before
    • The polymer nanoparticle composites presented in this paper has not been previously synthesized using APP. APP was used before to synthesize ZnO polymer nanoparticle composites (with methylmethacrylate and HDDA). In this work aerosol thiol-ene photopolymerization is used (utilizing thiol and ene monomers) to synthesize Ag-polymer nanoparticle hybrids with thiol-functonalities within the polymer for the first time. The challenges of using silver nanoparticles in this research are described in introdunction (lines 72-94).
  2. silver ions cytotoxicity.
    • This has not been tested, this will be targeted in future work  (added text, see 398-401).
  3. stabilization of silver
    • Silver nanoparticles are prone to aggregate in several systems, including organic solutions (such as the one we use for aerosol photopolymerization). Silver nanoparticles do not aggregate in initial solutions (for the commercially available silver nanoparticles used in our paper). The aggregation of silver nanoparticles is first observed in spray solutions during atomization. Therefore, a stabilizer (such as α-LA) was added to prevent the aggregation during aerosol generation.
  4. Line 55 Corrections applied (now lines 54-55)
  5. Line 78 Corrections applied (now lines 76-77)
  6. Line 129 Corrections applied (now lines 131-132)
  7. Line 141 Corrections applied (now lines 143, 150)
    • The energy used for the sonication was not recorded. The paper you are referring to is on fluid dynamics and there, the ultrasonic bath was designed specifically for CFD. We only use the bath to sonicate the dispersions to make them homogeneous. In this work we used a commercially available sonicator. The 100 mL flask (line 151) used for aerosol generation was placed in the center of the bath and sonicated at the set amplitude. 
  8. Section 2.4.
    • The operating parameters of the SEM are presented in lines 203-204, of the TEM in 212-213. TEM images were taken by means of a TVIPS camera F416 with a 4k x 4k CMOS chip. 

      For the SEM two different detectors were used: first detector (Signal A, ETD) is an "Everhart-Thornley-Detector", the second one is an In-Lens-detector (Signal B, InLens, with scintillator and photomultiplier), in this case only for the detection of secondary electrons, that means imaging of topography. For the 2000x magnification only the ETD was used, for other magnifications the signals of both detectors were used (50/50).

  9. Lines 269-271 Corrections applied (now lines 273-276)

  10. Lines 283-284.
    • No stabilizer was added when the sonication was used as a stabilization method to break down the silver nanoparticle aggregates formed during atomization.
  11. Lines 391-395.
    • New reference added (now lines 397-401). Please, see 

      https://doi.org/10.1046/j.1472-765X.1997.00219.x

      I quote: “compounds containing thiol groups reduce the toxicity of silver to Pseudomonas aeruginosa”.

Kind regards,

Authors
